# DIFFERENTIABLE NEURAL NETWORK ARCHITECTURE SEARCH

**Richard Shin**[*] **& Charles Packer**[*] **& Dawn Song**
University of California, Berkeley
{`ricshin,cpacker,dawnsong`}@berkeley.edu

## ABSTRACT

The successes of deep learning in recent years has been fueled by the development of innovative new neural network architectures. However, the design of a neural network architecture remains a difficult problem, requiring significant human expertise as well as computational resources. In this paper, we propose a method for transforming a discrete neural network architecture space into a continuous and differentiable form, which enables the use of standard gradient-based optimization techniques for this problem, and allows us to learn the architecture and the parameters simultaneously. We evaluate our methods on the Udacity steering angle prediction dataset, and show that our method can discover architectures with similar or better predictive accuracy but significantly fewer parameters and smaller computational cost.

## 1 INTRODUCTION

Deep neural networks have seen great success at solving problems in difficult application domains (speech recognition, machine translation, object recognition, motor control), and the design of new neural network architectures better suited to the problem at hand has served as a crucial source for these advancements. Several recent works have treated neural network architecture design as a reinforcement learning problem (Zoph & Le, 2016; Baker et al., 2016; Zoph et al., 2017). While these approaches have successfully found interesting architectures for highly-studied benchmark problems, these methods require training (tens of) thousands of models from scratch and testing them on a validation set.

In this paper, we treat network architecture search as a "fully differentiable" problem, and attempt to simultaneously find the architecture and the concrete parameters for the architecture that best solve a given problem. Unlike random, grid search, and reinforcement learning based search, we can obtain this result by training a single model which is roughly the same size as this maximal architecture, rather than needing to train a potentially very large number of different models to achieve a similar level of performance. By learning the architecture and the parameters simultaneously, we obtain a new design point with different trade-offs in the solution space of neural network architecture space.

Our approach has deep connections to network compression, where the goal is to take an existing neural network and reduce the number of parameters and the computational cost with minimal impact on the models' prediction accuracy. LeCun et al. (1989) and Hassibi et al. (1993) remove single weights from the network by examining second-order derivatives of the loss function, while Han et al. (2015) simply removed small weights and retrained the remaining weights. Setting logically contiguous blocks in the weight matrix to 0 allows for more significant speedups, and several methods use *group lasso* regularization to obtain this effect: Wen et al. (2016), Lebedev & Lempitsky (2016), Zhou et al. (2016) and Alvarez & Salzmann (2016). Li et al. (2016) remove feature maps from the weights of a convolutional kernel to achieve structured sparsity instead of group lasso. Liu et al. (2017) apply L1 regularization to the channel-wise scaling parameter in batch normalization for convolutional layers, and remove channels with corresponding parameters close to 0. The core regularization methods used in the paper build upon the prior work mentioned, but we have combined them in various ways, especially for learning grouped convolutions.

---

[*]Equal contribution.

## 2    OVERVIEW OF APPROACH FOR CONVOLUTIONAL NETS

While our overall approach is more general, for concreteness we only describe our approach for convolutional nets. For each convolutional layer, we would like to search over filter sizes, number of channels, and grouped convolutions. To handle the first two choices, we can replace each convolution with $\sum_{i,j=1}^{n,m} \alpha_i \beta_j (W^{(i,j)} * x)$, where $*$ is the convolution operator, $n, m$ are numbers of possible filter sizes and channel counts, and $\alpha_i, \beta_j$ are scalars that represent the strength of choice $i$, and $j$ (grouped convolutions are handled later). When only one of the $\alpha_i$ and $\beta_j$ are nonzero, then we have selected a filter size and number of channels for the layer.

As $nm$ may be very large, it is impractical to actually perform $nm$ convolutions and compute a weighted sum. Instead, we use the *linearity of convolution*: $\alpha(W * x) + \beta(V * x) = (\alpha W + \beta V) * x$, when $W$ and $V$ are of the same size. We now explain how to create $W^{(i,j)}$ for each filter size and number of channels so that they are all the same size and we can exploit linearity of convolution.

**Filter sizes.**    Let us consider a set of possible filter sizes $h_1 \times w_1, \cdots, h_n \times w_n$ that we wish to apply to an input. In order to ensure that the size of the output remains constant no matter the size of the filter, we note that we need to pad the input image by $h_i - 1$ pixels vertically and $w_i - 1$ pixels horizontally. Furthermore, convolution with filters of size $h_i \times w_i$ is identical to the same convolution when these filters are padded with zeros on the sides so that they are of size $h_{max} = \max(h_1, \cdots, h_n)$ times $w_{max} = \max(w_1, \cdots, w_n)$. Therefore, we can fix all filters to size $h_{max}$ and $w_{max}$ with appropriate padding.

**Number of channels.**    Instead of adjusting the number of output channels, we will instead vary which subset of the input channels the convolutional layer acts upon. Imagine that we would like to convolve a kernel $W \in \mathbb{R}^{h \times w \times c \times c_{out}}$ on $c$ channels of an input image with $c_m$ channels. We can transform this into a convolution with kernel $W' \in \mathbb{R}^{h \times w \times c_m \times c_{out}}$ where $W'_{\cdot, \cdot, k_i, \cdot} = W_{\cdot, \cdot, i, \cdot}$ for all $0 \leq i < c$ and $0 \leq k_i < c_m$, where none of the $k_i$ overlap, and all other parts of $W'$ are zero. Since convolution with $W'$ multiplies only $c$ channels in the image with the parts of $W'$ that are non-zero, the remaining $c_m - c$ channels have no effect on the ouput.

**Merging parameters.**    We can now have one convolution with a kernel of size $h_{max} \times w_{max} \times c_m \times c_{out}$, but this kernel is computed from $(\sum_{i=1}^n \sum_{j=1}^m h_i \times w_i \times c_j) \times c_{out}$ parameters. We add a further parameter sharing constraint across $W^{(i,j)}$ to reduce the number of parameters down to exactly $h_{max} \times w_{max} \times c_m \times c_{out}$. More specifically, we require $W'^{(i,j)}_{a,b,c,d} = W'^{(i',j')}_{a,b,c,d}$ for all $1 \leq i \leq n$ and $1 \leq j \neq m$ unless $W'^{(i,j)}_{a,b,c,d} = 0$ due to padding being placed at that particular location. We can then use a single parameter tensor of size $h_{max} \times w_{max} \times c_m \times c_{out}$, and scale each element with the sum of the relevant $\alpha_i$ to obtain the effective kernel.

**Grouped convolutions.**    In a grouped convolution, unlike a regular convolution, each output channel is connected to a subset of the input channels, to reduce the number of parameters and save computation compared to a full connectivity pattern. Grouped convolutions are used by architectures such as ResNeXt (He et al., 2017), Inception (Szegedy et al., 2016), and Xception (Chollet, 2016). To allow learning of grouped convolutions, we add the following architecture parameters to each convolutional layer: $P \in \mathbb{R}^{c_m \times p}$ and $S \in \mathbb{R}^{p \times c_{out}}$. We will then scale the combined $W_{\cdot, \cdot, i, j}$ with $(P \cdot S)_{i,j}$, for $0 \leq i < c_m$ and $0 \leq j < c_{out}$. Intuitively, each column of $P$ specifies a *connectivity pattern*: if a value in the column is 0, that means the corresponding input channel is ignored. $p$ is a hyper-parameter specifying the maximum number of connectivity patterns. Each row of $S$ specifies the assignment of patterns to an output channel; if the $k$th value in a row of $S$ is non-zero and the others are zero, the corresponding output channel is influenced only by the input channels prescribed in the $k$th column of $P$.

**Obtaining a discrete architecture.**    We cannot obtain a discrete architecture unless $\alpha_i, \beta_j$ are sparse; otherwise, representing the resulting network will require an architecture equivalent to the largest possible architecture in the space. To ensure this, we use $L_1$ regularization over $\alpha_i$ and $\beta_j$ so that they become sparse; we also used the sparsemax function (Martins & Astudillo, 2016) on $\beta_j$.

| Model | FLOPs | Params. | Train Err. | Test Err. |
|---|---|---|---|---|
| Baseline | 840M | 1.03M | 0.0247 | 0.0945 |
| *Indiv. channels* | | | | |
| DAS ($t = 10^{-3}, \gamma = 10^{-8}$) | 692M | 1.01M | 0.0124 | **0.0889** |
| DAS ($t = 10^{-2}, \gamma = 10^{-6}$) | 402M | 0.43M | 0.0194 | 0.0912 |
| DAS ($t = 10^{-1}, \gamma = 10^{-8}$) | **210M** | 0.54M | 0.0682 | 0.0955 |
| *Grouped convs.* $\gamma = 10^{-6}$ | | | | |
| DAS ($t = 10^{-4}, \eta = 3 \cdot 10^{-6}$) | 480M | 0.11M | 0.0318 | 0.0978 |
| DAS ($t = 10^{-3}, \eta = 3 \cdot 10^{-6}$) | 508M | 0.19M | 0.0318 | 0.0980 |
| DAS ($t = 10^{-2}, \eta = 3 \cdot 10^{-5}$) | 409M | **0.04M** | 0.0691 | 0.0996 |

Table 1: Performance of differentiable architecture search on the Udacity dataset. The lowest errors (RMSE), FLOPs and parameters are in boldface. For *individual channels*, regularization was applied to incoming channels individually. For *grouped convolutions*, we applied the method described in section 2. $t$ indicates the pruning threshold, $\gamma$ indicates the strength of the $L_1$ regularization, $\eta$ indicates the strength of the grouped convolution regularization applied to $P$ and $S$.

For the grouped convolution parameters $P$ and $S$, we used the $L_{1,2}$ and $L_{2,1}$ norms, also referred to as *group lasso* (Yuan & Lin, 2006) and *exclusive lasso* (Zhou et al., 2010), separately over the rows and columns, to promote that: 1) the columns of $P$ differ from each other in the locations of their non-zero entries, so that they are not redundant with each other; 2) the columns of $P$ are sparse, so that each connectivity pattern refers only to a few incoming channels; 3) each row of $S$ is sparse (ideally, contains only one non-zero entry), so that each outgoing channel only refers to one connectivity pattern; and 4) $S$ only refers to a small number of connectivity patterns.

Finally, we clamp all architecture parameters smaller than a threshold $t$ to 0. To recover any slight performance declines caused by the loss of a small number of parameters, we can optionally resume training of the network for a short period of time.

## 3 EXPERIMENTAL VALIDATION AND FUTURE WORK

Our experiments focused on convolutional neural networks that we discussed heavily in the previous section, using each of the mechanisms from section 2. We evaluate our approach on an end-to-end steering angle prediction task, using over 100,000 labelled video frames (from car-mounted cameras positioned at three different angles) released by Udacity[1] . The self-driving vehicle domain is particularly suited to studying the accuracy/size trade-off on CNNs due to computational requirements for cost-effective onboard GPUs.

We based our architecture space on a network recently introduced by Bojarski et al. (2016) for end-to-end steering prediction, which was shown to have good performance on real-world driving tasks. We allow the network to search over a variety of filter sizes and input channels: the first three layers can select from filter height/width $\{1, 2, 3, 4, 5\}$, the next two from $\{1, 2, 3\}$, while the fully-connected layers can only select the number of input channels. We also tried searching over grouped convolutions instead of simply selecting the input channels to use.

Table 1 shows the subsequent size and performance of the learned model extracted from the composite model (which is the same size as the baseline). We were able to see significant reductions in the number of parameters by learning grouped convolutions, at the cost of some increase in the test error. In particular, we can decrease the number of parameters by over tenfold compared to when we simply remove some of the incoming channels. The architectures learned contain a variety of filter shapes, in addition to having much smaller connectivity patterns. For example, the first bold entry (DAS ($t = 10^{-3}, \gamma = 10^{-8}$)) is a network with two $3 \times 3$ filters, one $4 \times 5$, one $5 \times 4$, and one $5 \times 5$. The second bold entry (DAS ($t = 10^{-1}, \gamma = 10^{-8}$)) is a network with two $2 \times 2$ filters, one $3 \times 3$, one $3 \times 5$, and one $5 \times 4$. Unlike existing methods, this approach does not rely upon training many different separate models to perform the search, since it can be integrated with learning of the regular model parameters. For future work, we plan to expanding the search space considered in our approach to include deeper networks and additional hand-designed architecture and cell types.

---

[1]Open Source Self-Driving Car project: `https://www.udacity.com/self-driving-car`

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
