# OpenReview forum: "Differentiable Neural Network Architecture Search"
_ICLR.cc/2018/Workshop — Accept_

### Official Review · AnonReviewer4 · 2018-03-06
**Straightforward extension of the existing methods.**

**Rating:** 4
**Confidence:** 3

**Review:**

In this paper, the authors proposed using sparse regularization techniques to prune redundant filters in CNN thereby optimizing the network structure.

[Clarity]
Good. The authors described the basic idea clearly.

[Quality, Originality]
I think the idea of pruning redundant components with sparse regularizations is a common technique in machine learning.
For example, in multiple kernel learning, sparse regularizations are used to choose kernels.
As authors mentioned in introduction, similar ideas are also used in the neural network literatures.
It is therefore not clear how the proposed method is novel compared to those existing methods.

[Significance]
The experimental evaluation is conducted only on the proposed method.
It is therefore not clear whether the proposed method is superior to the other existing methods in practice.
Because of this reason, I think the impact of this paper is low.

---

### Official Review · AnonReviewer1 · 2018-03-09
**Differentiable NAS**

**Rating:** 8
**Confidence:** 2

**Review:**

The authors aim to perform neural architecture search, but unlike most recent work, don't treat it as a reinforcement learning problem -- instead, they treat it as a "differentiable problem", where the architecture and parameters can be trained jointly. They use a convolutional neural network as an example and discuss how to optimize filter size and number of channels for each layer. They show their results on an experiment using the Udacity dataset.

I thought this was a good paper with novel contributions, as it looks at architecture search in a way that's different than most current methods. I would add that they should compare against more baselines such as standard architectures or other methods for architecture search to make the experiment stronger. Also, the number of layers seems like it has to be fixed -- how is this chosen in advance?

---

### Official Review · AnonReviewer2 · 2018-03-09
**Interesting idea but lack of explainations**

**Rating:** 6
**Confidence:** 3

**Review:**

This article tries to learn hyperparater during the main learning process, by using a global differentiable formulation. This paper belongs to the very attractive domain of meta-learning, which aims at making any ML algorithm to work at once, without a costly optimization of its structure/parameters. The authors explain that they focus on convnets and propose a group lasso optimization approch to optimize a group of filters. They also use large filters and rely on the regularization framwork to find the best actual filter size.
Doing this, they obtain good results on the udacity self driving car dataset using either a reduced number of parameters or a reduced computation time in the optimization process.

I do not catch the differentiable aspect of the architecture: filter are yet differentiable in classical CNN. I  do not get how padding max-sized filter with 0 make the size parameter differentiable.
More generally, what is the link between group lasso & differentiable architecture?
The authors claim that "we propose a method for transforming a discrete neural network architecture space into a continuous
and differentiable form". I do not see clearly the contribution in that sense.

The article is interesting but some key information is missing to understand the contribution.

---

### Decision · Program_Chairs · 2018-03-20
**ICLR 2018 Workshop Acceptance Decision**

**Decision:**

Accept

**Comment:**

Congratulations, your paper was accepted to the ICLR workshop.